# Mixture of Experts Made Intrinsically Interpretable

**Xingyi Yang** [1,2]  **Constantin Venhoff** [1]  **Ashkan Khakzar** [1]  **Christian Schroeder de Witt** [1]  **Puneet K. Dokania** [1]
**Adel Bibi** [1]  **Philip Torr** [1]

## Abstract

Neurons in large language models often exhibit *polysemanticity*, simultaneously encoding multiple unrelated concepts and obscuring interpretability. Instead of relying on post-hoc methods, we present **MoE-X**, a Mixture-of-Experts (MoE) language model designed to be *intrinsically* interpretable. Our approach is motivated by the observation that, in language models, wider networks with sparse activations are more likely to capture interpretable factors. However, directly training such large sparse networks is computationally prohibitive. MoE architectures offer a scalable alternative by activating only a subset of experts for any given input, inherently aligning with interpretability objectives. In MoE-X, we establish this connection by rewriting the MoE layer as an equivalent sparse, large MLP. This approach enables efficient scaling of the hidden size while maintaining sparsity. To further enhance interpretability, we enforce sparse activation within each expert and redesign the routing mechanism to prioritize experts with the highest activation sparsity. These designs ensure that only the most salient features are routed and processed by the experts. We evaluate MoE-X on chess and natural language tasks, showing that it achieves performance comparable to dense models while significantly improving interpretability. MoE-X achieves a perplexity better than GPT-2, with interpretability surpassing even sparse autoencoder (SAE)-based approaches.

## 1. Introduction

Transformer-based large language models (LLMs) (Radford et al., 2019; Brown et al., 2020; Waswani et al., 2017) have achieved remarkable progress. However, their internal

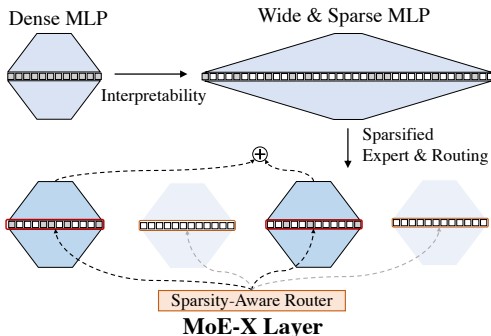

*Figure 1.* MoE-X introduces a sparse and wide network architecture designed for interpretability. Compared to dense MLPs, it incorporates both sparsity and a wider structure. Unlike traditional MoE models, it enforces sparsity within each expert and routes tokens to the sparsest experts.

workings remain poorly understood. This lack of understanding often leads to unexpected and potentially harmful behaviors (Hendrycks et al., 2023; Ngo et al., 2022), posing risks in their deployment. To address this, mechanistic interpretability (Elhage et al., 2022c) seeks to uncover how these models process information and reduce potential risks.

A central obstacle to mechanistic interpretability is *polysemanticity*, where individual neurons encode multiple, unrelated concepts (Olah et al., 2020). Specifically, we refer to the hidden neurons from the multi-layer perceptron (MLPs) in Transformers. Such polysemantic neurons lack clear, singular roles, making it difficult to identify disentangled features or factors in neural networks.

A common strategy to address this issue is to decompose entangled neuron activity into interpretable vectors using post-hoc methods like Sparse Auto-Encoders (SAEs) (Huben et al., 2023). However, these approaches are computationally expensive (Gao et al., 2024; Lieberum et al., 2024), require additional analysis after training, and often do not explain all the features of the model (Menon et al., 2024).

Instead, we advocate for designing interpretability *directly* into the model architecture in which the resultant discourages polysemanticity during training. While some works explore architectural changes for interpretability, they often

[1]University of Oxford [2]National University of Singapore.

*Proceedings of the 42ˢᵗ International Conference on Machine Learning*, Vancouver, Canada. PMLR 267, 2025. Copyright 2025 by the author(s).

focus on toy-scale tasks (Pearce et al., 2024; Agarwal et al., 2021; Sharkey, 2023; Jermyn et al., 2022) or compromise performance (Elhage et al., 2022a).

To achieve built-in interpretability, as shown in Figure 1, we identify two key factors that influence it, (1) increasing the size of the MLP layers, i.e., number of hidden activations, and (2) increasing the sparsity of these activations. That is to say, making the MLP layers in a transformer *wider* and *sparser* should encourage more disentangled internal representations. To test this beyond toy experiments (Jermyn et al., 2022; Elhage et al., 2022b), we conduct experiments on GPT-2-like (Radford et al., 2019) models on chess gameplay data (Karvonen et al., 2024). Chess provides an excellent natural "ground truth" for interpretability because the board state can be used as a reference for understanding how each neuron represents and predicts chess moves. Our experiments show that a sufficiently wide-and-sparse MLP in transformer indeed yields more interpretable neurons, leading to a 25% increase in F1 score for chess move prediction, supporting our hypothesis.

Motivated by these findings, we propose to leverage mixture-of-experts (MoE) architectures (Fedus et al., 2022; Shazeer et al., 2017) as a natural fit for *intrinsic interpretability*. While different from a standard MLP, MoE can be rewritten as a wide and sparse MLP, whose neurons are each expert's neuron weighted by the sparse gating scores. This structure allows MoE to increase the width while maintaining controlled sparsity, leading to inherently interpretable models.

Yet, typical MoE models are not perfect for interpretability. First, each expert is still a dense MLP, which can suffer from polysemanticity. Second, standard *top-k* routing (Fedus et al., 2022) primarily targets performance rather than expert sparsity. As a result, the gating decisions made by the routing mechanism are often misaligned with the goals of interpretability.

To bridge this gap and align MoE with interpretability, we propose **MoE-X**, which includes two key designs:

**ReLU Experts.** We use `ReLU` activation within each expert. This simple yet effective modification promotes intrinsic activation sparsity in the expert (Awasthi et al., 2024), which in turn helps to disentangle the feature representations.

**Sparsity-Aware Routing.** We introduce a gating function that predicts which experts would produce the most sparse activations. To avoid expensive computations, we develop a method to estimate each expert's sparsity *without* explicitly computing all activations. This ensures that the sparsest experts are chosen during inference, promoting disentangled representations.

Together, these modifications enable MoE-X to maintain competitive performance while providing more transparent,

semantically meaningful internal representations. Our experiments on chess and language tasks confirm that MoE-X matches or exceeds the performance of dense transformers while eliminating the need for expensive post-hoc interpretability methods.

In summary, our main contributions are:

a) We systematically analyze how architectural choices—particularly width and sparsity—influence interpretability in transformer-based language models.

b) We introduce MoE-X, a redesigned MoE layer that functions as a wide, sparse, and more interpretable MLP within large language models.

c) We incorporate *ReLU Experts* and *Sparsity-Aware Routing* to tightly connect gating decisions with the expected sparsity of activations.

d) Our experiments on chess tasks demonstrate that MoE-X models achieve strong performance while offering clear and interpretable representations.

## 2. Related Work

**Mechanistic Interpretability & Polysemantics.** Mechanistic interpretability (Olah, 2022) aims to understand deep neural networks by analyzing individual units (e.g., neurons) reverse-engineering their computations (Elhage et al., 2021). Although this approach provides insights into large language models (LLMs) (Zhong et al., 2024; Wang et al., 2022), many neurons remain polysemantic, activating for multiple concepts (Elhage et al., 2022b), making interpretation difficult. Post-hoc methods like Sparse Auto-Encoders (SAEs)(Gao et al., 2024; Huben et al., 2023) attempt to address this but are computationally expensive and incomplete (Menon et al., 2024). In this paper, we introduce a MoE architecture to reduce polysemanticity. This approach promotes more interpretable internal representations without the need for extensive post-hoc methods.

**Intrinsic Interpretability.** Intrinsic interpretability aims to design neural networks that are inherently easier to understand without sacrificing performance. These methods enforce sparsity, modularity, and monosemanticity through architectural and training constraints. For example, (Liu et al., 2023a;b) use brain-inspired modular training to enhance anatomical modularity in RNNs, while (Jermyn et al., 2022; Elhage et al., 2022a) explore structural choices for monosemanticity, and (Sharkey, 2023) employs bilinear layers for interpretability. In language modeling, some approaches incorporate concepts like bag-of-words (Hewitt et al., 2023) or utilize prototypes (Xie et al., 2023) to achieve this. We take a different approach by leveraging MoE for intrinsic interpretability.

**Mixture of Experts.** Mixture-of-Experts (MoE) models dynamically route input to specialized "experts" to reduce computation (Jacobs et al., 1991; Shazeer et al., 2017). Recent work focus on replacing MLP layers in LLMs with MoE layers, achieving better performance at lower cost (Jiang et al., 2024; Fedus et al., 2022). A major challenge in MoE is designing the routing function (Zhou et al., 2022), which typically requires an auxiliary loss to balance expert usage. Regarding interpretability, previous studies observed that MoE models tend to exhibit increased monosemanticity (Park et al., 2024; Oldfield et al., 2024). but these studies offer limited explanations for why this occurs. In this work, we clarify its underlying mechanisms and propose a redesigned routing function that prioritizes experts with more interpretable activations, rather than focusing solely on performance.

## 3. Preliminary Study: What Architectural Choices Enhance Interpretability?

To design more interpretable architectures, it is essential to identify what is the key influencing factors.

In this section, we conduct a series of toy experiments by training LLMs on chess gameplay data and evaluating their interpretability. Through extensive ablations of various design choices, we identify two key factors that significantly enhance inherent interpretability in language models:

a) **MLP Hidden Size**: Larger hidden states result in better interpretability.

b) **Sparsity of Hidden Activations**: Lower numbers of non-zero neurons lead to more interpretable representations.

In the next section, we will use these findings to design intrinsic interpretable architectures.

### 3.1. Measuring interpretability on Chess Games

Designing and evaluating the interpretability of language models is challenging due to the absence of a universal metric. In our experiments, we use a chess game dataset to assess interpretability (McGrath et al., 2022; Toshniwal et al., 2022; He et al., 2024). Specifically, we measure how well the model's internal activations align with semantically meaningful chess board state properties (BSP), using the metrics described in (Karvonen et al., 2024).

**Dataset and Metrics.** As shown in Figure 2, we trained LLMs on chess Portable Game Notation (PGN), treating it as a language, and analyzed the interpretability of MLP hidden activations. Specifically, we trained an 8-layer GPT2-like model (Radford et al., 2019). Each character in the PGN is treated as a token, and we conducted next-token prediction training.

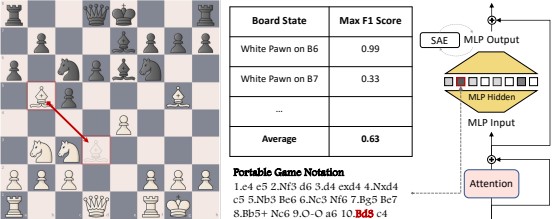

*Figure 2.* Illustration of using chess game to evaluate the LLM's interpretability.

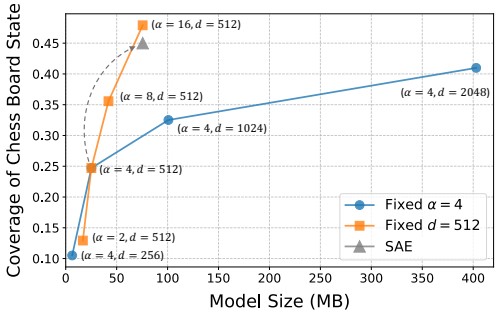

*Figure 3.* Comparision BSP Coverage score *v.s.* the Model size.

After training, we calculated the *BSP Coverage* Score on layer-6 MLP hidden activation. This score measures the average best F1 score when each feature is used as a classifier for BSPs. More information can be found in Section 5.1 and Appendix. We report the core observations on model hidden size and activation sparsity.

**Study I: Model Hidden Size.** In this study, we train models with different MLP hidden widths $D$. As shown in Figure 2, the width is determined by two factors: the input dimension of the MLP as $d$, and the hidden size multiplier $\alpha$. Together, the hidden size is $D = \alpha d$. We vary either $d$ or $\alpha$. With $\alpha = 4$ fixed, we test $D \in \{256, 512, 1024, 2048\}$. With $d = 512$ fixed, we test $\alpha \in \{2, 4, 8, 16\}$. Figure 3 compares BSP coverage for fixed $d$ and fixed $\alpha$, with model size on the x-axis. As a baseline, we plot the SAE score with a dictionary size of 4096, trained on `post-res` for model ($\alpha = 4, d = 512$). This results in nearly a 3× increase in model parameters.

We make three key observations. **First**, increasing both $d$ and $\alpha$ improves the interpretability, as indicated by a higher coverage score. **Second**, scaling $\alpha$ is better than scaling $d$. This implies that, instead of increasing the overall model size, we can efficiently scale the hidden size multiplier $\alpha$ for better interpretability. **Third**, by increasing the model size, we can eventually outperform the SAE features in terms of interpretability. For example, model with ($\alpha = 16, d = 512$) achieves a score of 0.53, compared to 0.45 for ($\alpha = 4, d = 512$)+SAE, with a similar overall parameter.

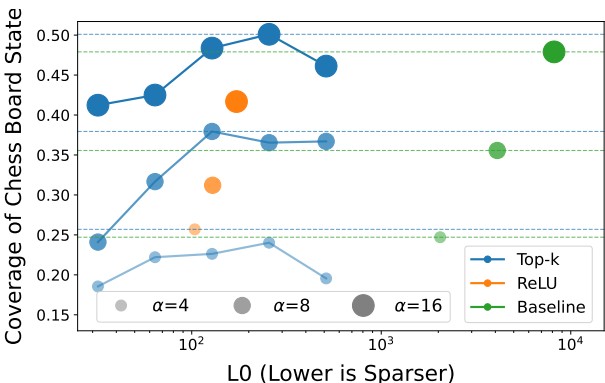

*Figure 4.* Comparing BSP Coverage score *v.s.* $L$-0 norm of the hidden.

**Study II: Activation Sparsity.** In this study, we explore how the activation sparsity affects the model interpretability. we explore different techniques to control the sparsity of the hidden activations in the MLP. Specifically, we either use ReLU activation instead of GELU, or apply Top-k activation (Makhzani & Frey, 2013; Gao et al., 2024)[1].

We visualize the results in Figure 4. We find that, for various model sizes, imposing sparsity typically improves coverage scores (difference between the dashed blue and green line). Moreover, these gains become more pronounced in larger models. However, the most extreme sparsity does not always yield the best interpretability, and identifying the optimal sparsity level remains difficult.

**Analysis and Understanding.** We hypothize wide and sparse neural networks are more interpretable as they minimize feature superposition (Elhage et al., 2022b). Width provides sufficient capacity for the model to assign distinct neurons to specific features. Sparse activations ensure that only a small subset of relevent neurons is active for a given input, which reduces interference between features.

Although similar architectural properties have been explored in non-language tasks and non-transformer architecture (Jermyn et al., 2022; Elhage et al., 2022b), or SAE features (Gao et al., 2024), evaluating those properties on language model pretraining was previously unexplored.

# 4. Mixture of Experts for Intrinsic Interpretability

Building on the observations in Section 3, we aim to design architectures that are both wide and sparsely activated, ensuring better interpretability. Sparse Mixture of Experts (SMoE) naturally aligns with these properties, making it

---

[1]We also experimented with $\ell_1$ regularization but found it ineffective for reducing the $\ell_0$ norm and challenging to tune, so we excluded it.

an excellent candidate for such a design. In this section, we first explain how SMoE works and demonstrate how it aligns with wide-and-sparse principles. Building on top of this, we propose a new MoE layer designed to further enhance interpretability.

## 4.1. Preliminary: Sparse Mixture of Experts

SMoE improves efficiency by activating only a subset of computations per input. Unlike traditional models that use a single MLP for each LLM layer, SMoE employs multiple parallel MLPs, referred to as "experts". Each token is routed to a subset of experts, and their predictions are combined using weights determined dynamically based on the input.

Formally, SMoE comprises two key components: experts and a gating network.

- **Experts:** Let $\mathbf{x} \in \mathbb{R}^d$ be the input vector of one token. A SMoE layer consists of $M$ experts, each represented as $f_j(\mathbf{x})$ for $j \in \{1, 2, \ldots, M\}$. Typically, each expert is a small MLP

$$f_j(\mathbf{x}; \theta_j) = \mathbf{W}_{\text{dec}}^{(j)} \mathbf{z}^{(j)}, \mathbf{z}^{(j)} = \sigma\left(\mathbf{W}_{\text{enc}}^{(j)} \mathbf{x}\right) = \sigma\left(\mathbf{h}^{(j)}\right),$$

where $\mathbf{W}_{\text{dec}}^{(j)} \in \mathbb{R}^{d \times D}$ and $\mathbf{W}_{\text{enc}}^{(j)} \in \mathbb{R}^{D \times d}$ are weight. $\mathbf{h}^{(j)}$ and $\mathbf{z}^{(j)}$ are pre-activation and post-activation vectors. Here, $D$ represents the hidden dimension of each expert.

- **Gating Network:** The gating network $g(\mathbf{x}; \phi)$ generates a set of weights $\mathbf{w} = [\omega_1, \omega_2, \ldots, \omega_M] \in \mathbb{R}^M$, with each $\omega_j$ indicating the contribution of expert $j$ to the output. These weights are computed using a learnable matrix $\mathbf{W}_g \in \mathbb{R}^{M \times d}$ as follows:

$$\mathbf{w} = g(\mathbf{x}; \phi) = \text{Softmax}\left(\text{TopK}\left(\mathbf{W}_g \mathbf{x}\right)\right), \quad (1)$$

Only top-k values are retained and normalized using softmax (Shazeer et al., 2017), while the rest are set to zero.

The final output of the MoE model, $\hat{\mathbf{y}}$, is a weighted sum of the expert outputs $\hat{\mathbf{y}} = \sum_{j=1}^{M} \omega_j f_j(\mathbf{x}; \theta_j)$. Since most $\omega_j$ are zero, this model is referred to as a "Sparse" MoE.

## 4.2. SMoE is a Natural Fit for Interpretability

SMoE naturally aligns with our identified interpretable architecture, as it is both wide and sparsely activated. To see this, consider a "mega-decoder" by concatenating all expert decoder matrices $\mathbf{W}_{\text{dec}} = \text{concat}([\mathbf{W}_{\text{dec}}^{(1)}, \ldots, \mathbf{W}_{\text{dec}}^{(M)}]) \in \mathbb{R}^{d \times MD}$. We can also define a new hidden code as $\mathbf{z} = \text{concat}([\omega_1 \mathbf{z}^{(1)}, \ldots, \omega_M \mathbf{z}^{(M)}]) \in \mathbb{R}^{MD}$. Here, each $\omega_j \mathbf{z}^{(j)}$ is the scaled activation from $j$-th expert. Notably, decoding $\mathbf{z}$ through $\mathbf{W}_{\text{dec}}$ exactly the same as SMoE output (Liu et al.,

2023c)

$$\hat{\mathbf{y}} = \sum_{j=1}^{M} \omega_j f_j(\mathbf{x}; \theta_j) = \sum_{i=1}^{M} \mathbf{W}_{\text{dec}}^{(j)} \left( \omega_j \mathbf{z}^{(j)} \right) = \mathbf{W}_{\text{dec}} \mathbf{z},$$

(2)

In other words, SMoE acts like a larger MLP whose hidden layer is each expert's activations, scaled by its gating score. Because only top-k non-zero $\omega_j$ are retrained, $\mathbf{z}$ is *structured sparse*. When $\omega_j = 0$, all elements in $\omega_j \mathbf{z}^{(j)}$ are zero. Consequently, SMoE is "wide", as its hidden dimension is $MD$, but also "sparse", as activations are restricted to a subset of experts. In this way, SMoE satisfies the criteria of wide and sparsely activated MLP that supports interpretability.

**SMoE is not perfect for interpretability.** Despite its inherent sparsity and modularity, two key issues remain. **First**, activations in each expert are still dense, which can still lead to polysemantic features. **Second**, the gating function is trained purely for performance, so its values may not reflect interpretable expert properties. We address these issues in the following sections.

### 4.3. Designing SMoE for Greater Interpretability

To address the issues discussed above, we redesign both the expert architecture and the routing function to enforce neuron-level sparsity within each expert.

#### 4.3.1. RELU EXPERT FOR ACTIVATION SPARSITY

To address the first challenge, we adopt the `ReLU` function as the non-linear activation $\sigma(\cdot)$ for each expert MLP. Empirically, we observe that experts trained with `ReLU` exhibit a high degree of activation sparsity, which helps disentangle features while maintaining strong performance. While intrinsic activation sparsity has been studied in the context of efficiency (Zhang et al., 2024; Mirzadeh et al., 2023; Awasthi et al., 2024), its role in enhancing interpretability is less explored.

#### 4.3.2. SPARSITY-AWARE ROUTING

To tackle the second issue, we aim to route each input token to the expert $f_j$, which produces *sparsest activation* (i.e., fewest non-zero entries). Formally, if $\mathbf{z}^{(j)} = \texttt{ReLU}(\mathbf{h}^{(j)})$, then the sparsity of $\mathbf{z}^{(j)}$ can be discribed using its $\ell_0$-norm

$$\|\mathbf{z}^{(j)}\|_0 = \sum_i \mathbb{I}(\mathbf{h}_i^{(j)} \geq 0),$$

(3)

where $\mathbb{I}(\cdot)$ stands for the indicator function. The simplest approach to do gating is to evaluate $\mathbf{z}^{(j)}$ for all experts and select the one with the fewest positive elements. However, this contradicts the SMoE principle of limiting computation to a subset of experts. Instead, we use a cheap proxy

for $\|\mathbf{z}^{(j)}\|_0$ based on probabilistic assumptions about the encoder weights $\mathbf{W}_{\text{enc}}^{(j)}$.

**Approximate Sparsity via Gaussian Assumptions.** For expert $j$, assume each column of the encoder weight matrix $\mathbf{W}_{\text{enc}}^{(j)}$ is drawn i.i.d. from the same Gaussian distribution, $\{w_{m,i}\}_{m=1}^{D} \sim \mathcal{N}(\mu_i^{(j)}, (\sigma_i^{(j)})^2)$. Then each component $h_i^{(j)}$ of the pre-activation vector $\mathbf{h}^{(j)}$ is a sum of Gaussian random variables and thus also follows a Gaussian distribution:

$$h_i^{(j)} \sim \mathcal{N}\left( \mu_h^{(j)}, (\sigma_h^{(j)})^2 \right) = \mathcal{N}\left( \sum_i \mu_i^{(j)} x_i, \sum_i (\sigma_i^{(j)})^2 x_i^2 \right),$$

(4)

Thus, the probability that $h_i^{(j)}$ is positive is given as $\text{Prob}(h_i^{(j)} > 0) = \Phi\left( \mu_h^{(j)} / \sigma_h^{(j)} \right)$ where $\Phi(h)$ is the CDF of the normal distribution. Since each elements in $\mathbf{h}^{(j)}$ follows the same distribution, the $\ell_0$-norm could be estimated as the expected value of

$$\|\mathbf{z}^{(j)}\|_0 \approx \sum_i \mathbb{E}[\mathbb{I}(\mathbf{h}_i^{(j)} \geq 0)] = D\Phi\left( \frac{\mu_h^{(j)}}{\sigma_h^{(j)}} \right). \quad (5)$$

Hence, we pick expert(s) that minimize $\|\mathbf{z}^{(j)}\|_0$, i.e., those with the lowest probability of being positive.

**Router Implementation.** In practice, we estimate $\mu_h^{(j)}$ and $\sigma_h^{(j)}$ efficiently using column-wise statistics of $\mathbf{W}_{\text{enc}}^{(j)}$ and $\mathbf{x}$. That is to say, for expert $j$

$$\boldsymbol{\mu}^{(j)} = \frac{1}{D} \sum_{m=1}^{D} \mathbf{W}_{\text{enc}}^{(j)}[m, :]$$

$$\boldsymbol{\sigma}^{(j)} = \frac{1}{D} \sum_{m=1}^{D} \left( \mathbf{W}_{\text{enc}}^{(j)}[m, :] - \boldsymbol{\mu}_h^{(j)} \right)^2,$$

where we have that $\mu_h^{(j)} = \boldsymbol{\mu}^{(j)\top} \mathbf{x}$ and $\sigma_h^{(j)} = \boldsymbol{\sigma}^{(j)\top}(\mathbf{x}^2)$. Using the CDF approximation $\Phi(h) \approx \frac{1}{2}(1 + \text{erf}(h/\sqrt{2}))$, we compute the gating weights via a `TopK+Softmax` operation

$$\mathbf{w} = g(\mathbf{x}; \phi) = \text{Softmax}\left( \text{TopK}\left( -\text{erf}\left( \frac{\mu_h^{(j)}}{\sqrt{2}\sigma_h^{(j)}} \right) \right) \right).$$

(6)

Here, $\text{erf}(\cdot)$ is the error function. The resulting gating scores select the experts and provide an inexpensive estimate of each expert's activation sparsity.

**Routing Regularizes Sparsity.** Since we do not `detach` the gradient of $\mathbf{W}_{\text{enc}}^{(j)}$, the gating function implicitly regularizes the experts to produce sparser activations. Specifically, when the model learns to favor an expert $j$ by increasing its gating score $\omega_j$, it simultaneously updates $\mathbf{W}_{\text{enc}}^{(j)}$, to encourage activation $\mathbf{z}^{(j)}$ to be sparser .

*Table 1.* Comparison with baseline method by keeping model activated parameters the same.

| Model | Val Loss↓ | Coverage↑ | Reconstruction↑ |
|---|---|---|---|
| GELU (GPT-2) | 0.213 | 0.356 | 0.608 |
| *Activation Function* | | | |
| ReLU | 0.215 | 0.312 | 0.581 |
| GEGLU | 0.209 | 0.255 | 0.394 |
| SoLU | 0.216 | 0.306 | 0.343 |
| *Mixture-of-Experts* | | | |
| Monet-HD | **0.210** | 0.312 | 0.528 |
| Monet-VD | 0.212 | 0.283 | 0.482 |
| PEER | 0.214 | 0.323 | 0.426 |
| Switch | 0.212 | 0.424 | 0.734 |
| MoE-X | 0.211 | **0.428** | **0.840** |

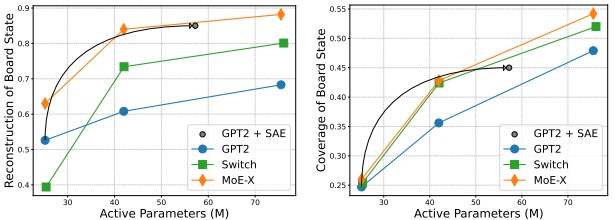

*Figure 5.* BSP Coverage and Reconstruction score of different model sizes.

**Computational Complexity.** Given input $\mathbf{x} \in \mathbb{R}^{N \times d}$, standard *top-k* gating has complexity $\mathcal{O}(NMd)$. Naively computing all activations and their $\ell_0$-norm costs $\mathcal{O}(NMDd)$. In contrast, our Sparsity-Aware Routing only requires $\mathcal{O}(MDd)$ for column-wise statistics plus and $\mathcal{O}(NMd)$ for inner products, resulting in a total complexity of $\mathcal{O}((N + D)Md)$. This design scales efficiently with large numbers of experts and high-dimensional inputs.

# 5. Experiments

In this section, we conduct experiments on the chess and language datasets to validate the design of MoE-X, focusing on both performance and interpretability.

## 5.1. Chess Play Experiments

**Experimental Setup.** For chess experiments, we train models on `lichess_6gb`[2] (Karvonen, 2024), a 16 million games from the public Lichess chess games database. The input to the model is a chess PGN string (`1.e4 e5 2.Nf3 ...`) of a maximum length of 1023 characters, with each character representing an input token. The model's vocabulary consists of the 32 characters necessary to construct chess PGN strings. We split the dataset into 99% of training corporse and validation on 1% of validation set, report validation loss to test performance. Addition-

[2]https://huggingface.co/datasets/adamkarvonen/chess_games

ally, we report the *BSP Coverage and Reconstruction* score defined in (Karvonen et al., 2024) to assess interpretability.

We compared our proposed MoE-X against three families of models. The first is a dense baseline model similar to GPT-2. The second includes models with activation functions designed for better interpretability, such as bilinear layers like GEGLU (Pearce et al., 2024; Shazeer, 2020) and SoLU (Elhage et al., 2022a). The third consists of MoE models, including fine-grained MoEs like Monet (Park et al., 2024) and PEER (He, 2024), as well as standard MoEs like the Switch Transformer (Fedus et al., 2022). For Switch Transformer and MoE-X, we use 8 experts, with 2 experts activated at a time. For MoE models, we explain the scaled hidden representation $\mathbf{z}$ defined in Section 4.2. This avoids using the raw activations from each expert.

All models have 8 layers and are trained for 60k iterations with a batch size of 100. We use the AdamW optimizer with an initial learning rate of 3e-4 and cosine scheduling to reduce the learning rate to 1e-4 in the end. We train MoE-X by upcycling the weights (Komatsuzaki et al., 2022) from dense model. Additionally, we applied a load balance loss with a value of $\lambda = 0.001$. All experiments were conducted on 4 NVIDIA A40 GPUs. More details are listed in Appendix.

**MoE Achieves Better Interpretability.** We present the interpretability scores of different models in Table 1. To ensure a fair comparison, we *strictly match the number of activated parameters* between dense models and MoE models. For dense models, we use an MLP hidden size of $D = 4096$, while for MoE models, we activate 2 experts, each with 2048 hidden neurons.

Several key observations emerge from the results. First, MoE models demonstrate superior interpretability. Swicth transformer readily improve the interpretability score, and our proposed MoE-X achieves the best *Reconstruction Score* of 0.84.

Second, prior architecture designs claiming improved interpretability do not perform well in practice. For example, SoLU's scores is even lower than the GELU-based GPT-2 baseline. Similarly, recent MoE models claiming to improve monosemanticity, such as Monet (Park et al., 2024), do not do well. We hypothesize that this is due to the use of product key quantization (Lample et al., 2019), which relies on the Cartesian product. This method makes expert gating scores interdependent, preventing experts from functioning independently—a key requirement for interpretability. These findings call for a thorough re-evaluation of this field, as many claims of improved interpretability lack strong empirical support.

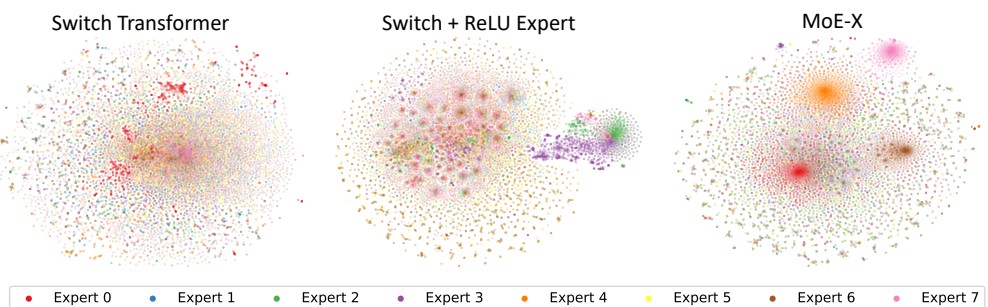

Switch Transformer     Switch + ReLU Expert     MoE-X

● Expert 0   ● Expert 1   ● Expert 2   ● Expert 3   ● Expert 4   ● Expert 5   ● Expert 6   ● Expert 7

*Figure 6.* t-SNE projections of encoder weights for original MoE layer, MoE with ReLU experts, and without full MoE-X layers, trained on Chess dataset.

**MoE-X Scales Interpretability Faster.** We evaluate interpretability across different model sizes, comparing dense GPT-2, Switch Transformers, and MoE-X. To compare the size fairly, for dense models, we fix $D = 512$ and vary $\alpha \in \{4, 8, 16\}$. For MoEs, we set $D = 512$ and $\alpha = 4$ for each expert, while varying the number of activated experts $k \in \{1, 2, 4\}$.

As shown in Figure 5, interpretability scores improve significantly as model size increases. With the same number of activated parameters during inference, MoE-X consistently outperforms alternatives, particularly in the *BSP Reconstruction Score*.

**MoE-X beats SAE with Greater Faithfulness.** We compare MoE-X with SAE trained on GPT-2-small `post-res` with a SAE hidden size of 4096. As shown in Figure 5, MoE-X achieves better interpretability than SAE with the same total parameters (GPT-2 + SAE).

Moreover, MoE-X is inherently more faithful due to its intrinsic interpretability. Unlike SAE, which relies on post-hoc decomposition to approximate features, MoE-X directly learns interpretable features. As a result, SAE always suffers some performance loss ($\sim 96\%$ validation loss), while MoE-X achieves perfect fidelity ($100\%$ loss recovery).

**MoE-X Expert cluster features.** To better understand the models, we visualize the encoder weights of different MoE models trained on chess data using t-SNE (Van der Maaten & Hinton, 2008) projections. We treat each row from $\mathbf{W}_{enc}^{(j)}$ is treated as a data point, and apply t-SNE to project them onto a 2D plot. As shown in Figure 6, our MoE-X effectively clusters vectors for expert $0, 4, 6, 7$, capturing topics related to interpretable factors. In contrast, vanilla MoE models, such as the Switch Transformer, use routing functions optimized solely for performance, which fail to form meaningful cluster of features.

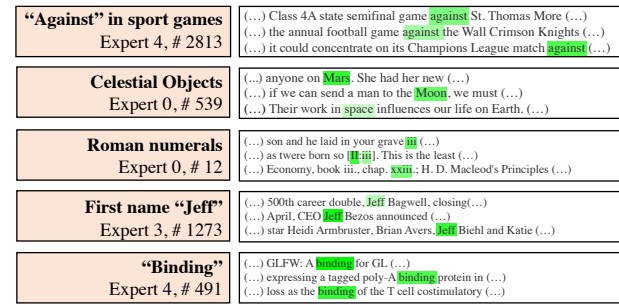

| "Against" in sport games
Expert 4, # 2813 | (…) Class 4A state semifinal game against St. Thomas More (…)
(…) the annual football game against the Wall Crimson Knights (…)
(…) it could concentrate on its Champions League match against (…) |
| Celestial Objects
Expert 0, # 539 | (…) anyone on Mars. She had her new (…)
(…) if we can send a man to the Moon, we must (…)
(…) Their work in space influences our life on Earth. (…) |
| Roman numerals
Expert 0, # 12 | (…) son and he laid in your grave iii (…)
(…) as twere born so [ii·iii]. This is the least (…)
(…) Economy, book iii., chap. xxii ; H. D. Macleod's Principles (…) |
| First name "Jeff"
Expert 3, # 1273 | (…) 500th career double, Jeff Bagwell, closing(…)
(…) April, CEO Jeff Bezos announced (…)
(…) star Heidi Armbruster, Brian Avers, Jeff Biehl and Katie (…) |
| "Binding"
Expert 4, # 491 | (…) GLFW: A binding for GL (…)
(…) expressing a tagged poly-A binding protein in (…)
(…) loss as the binding of the T cell costimulatory (…) |

*Figure 7.* Activated tokens for experts in MoE-X small on RedPajama-v2 validation dataset. Their interpretations were identified using the auto-interpretation.

## 5.2. Interpretability for Natural Language

**Experimental Setup:** For natural language models, we pretrain on the 10BT subset of FineWeb (Penedo et al., 2024). We use a batch size of 320, a context length of 1024 tokens per sentence, and train all models for 100k gradient steps. We evaluate the models on OpenWebText (Gokaslan et al., 2019), LAMBADA (Paperno et al., 2016), WikiText103, and WikiText2 (Merity et al., 2016), and reported the perplexity (PPL) score to show the performance.

In addition to performance evaluation, we measure the interpretability by running the auto-interpretability pipeline and report the *Detection Accuracy*[3] defined in (Paulo et al., 2024). To obtain this score, we collect the activations of the target MLP over 10M tokens from RedPajama-v2 (Weber et al., 2024). The activated contexts are then fed into an explainer LLM, which provides a short interpretation for the corresponding neuron. A scorer LLM is asked to do a binary classification to determine whether a whole sequence activated a hidden neuron given an interpretation and test text. We report the accuracy of this classification. We use Llama 3.1b 70b instruct as both the scorer and the explainer model. More detail is in Appendix.

---

[3]https://github.com/EleutherAI/sae-auto-interp

*Table 2.* Language modeling performance for different architectures. For PPL, lower is better.

| Model | OpenWeb (PPL)↓ | LAMBADA (PPL)↓ | WikiText103 (PPL)↓ | WikiText2 (PPL)↓ |
|---|---|---|---|---|
| GPT-2 Small | 22.83 | 32.71 | 49.89 | 44.36 |
| GPT-2 Small w SAE | 31.60 | 38.21 | 55.33 | 49.16 |
| Switch-S (8×124M) | **18.36** | **27.63** | 45.22 | **38.90** |
| MoE-X-S (8×124M) | 19.42 | 28.11 | **43.80** | 42.58 |
| GPT-2 Medium | 17.19 | 24.31 | 37.87 | 35.70 |
| Switch-M (8×354M) | 15.43 | **20.82** | 35.41 | **34.71** |
| MoE-X-M (8×354M) | **14.78** | 21.34 | **35.01** | 35.16 |

We compare MoE-X with GPT-2 and Switch-Transformer. For GPT-2, we trained small (124M) and medium (354M) models. Similarly, for Switch-Transformer and MoE-X, we created small and medium configurations with 8 experts each. During inference, 2 experts are active, resulting in 180M and 555M active parameters for small and medium models. We evaluate interpretability at layer 8 for small models and layer 16 for medium models. GPT-2 with SAE is also evaluated at `post-res` of layer 8.

**Quantitative Experiments.** Table 2 presents the language modeling performance for different models. We observe that MoE models outperform dense model like GPT-2, with Switch Transformer slightly ahead of MoE-X but at a comparable level. Notably, GPT-2 performance drop significantly when running with SAE. This is because post-hoc explanations like SAE simply fail to capture all crucial features. It results in reduced performance and less faithful explanations.

For interpretability, we report the *Detection* Score for 1,000 randomly selected features. Each feature is scored with 100 activating and 100 non-activating examples. The activating examples are chosen via stratified sampling such that there are always 10 examples from each of the 10 deciles of the activation distribution. Figure 8 illustrates the overall accuracy. GPT+SAE serves as a strong baseline for interpretability, which MoE-X Small already matches. When the model size is increased to MoE-X Medium, interpretability improves further, surpassing SAE.

**Qualitative Experiments.** We show some auto-interp results on MoE-X small and its top-activated context in Figure 7. More results are included in Appendix 5. MoE-X successfully identifies interpretable concepts.

### 5.3. Ablation Study and Analysis

**ReLU Expert.** We verify the benefit of use ReLU experts by replacing it with default GELU function, and train on the chess dataset using a 2-of-8 expert setup. Since ReLU zeros out negative values while GELU does not, this replacement increases the average $\ell_0$ norm of hidden activations from $\sim 166$ to $\sim 4091$. Besides, as shown in Table 3 applying ReLU significantly improves the reconstruction score. Those experiments show that applying ReLU induces sparser and more interpretable features.

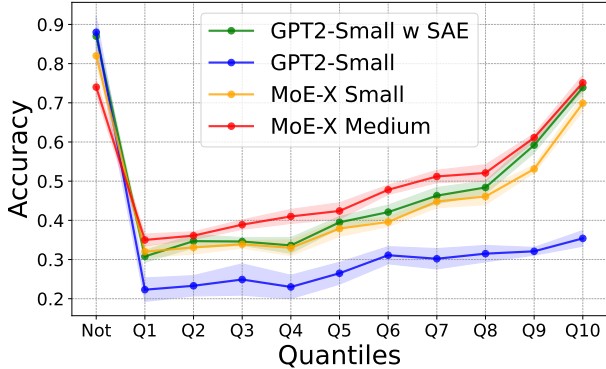

*Figure 8.* Automated Interpretability Detection Results in 8th Layer Hidden Activation Quantiles 1000 Random Features with 95% Confidence Intervals. **Not** indicates non-activating text.

*Table 3.* Ablation study of Routing and Expert Choice.

| ReLU Expert | Sparsity Router | Coverage | Reconstruction |
|---|---|---|---|
| ✗ | ✗ | 0.424 | 0.734 |
| ✗ | ✓ | 0.404 | 0.740 |
| ✓ | ✗ | 0.418 | 0.829 |
| ✓ | ✓ | **0.428** | **0.840** |

**Sparsity-Aware Routing.** We evaluate our gating function from two perspectives: (1) whether it selects sparser experts, and (2) whether it improves interpretability. Figure 9 shows the $\ell_0$-norm of each expert's activations across 5,000 sentences, alongside gating scores from both a standard top-k approach and our sparsity-based method. While standard gating often misestimates sparsity, our gating scores exhibit a strong negative correlation ($r < -0.95$) with the actual expert sparsity and consistently selects sparser experts. As shown in Table 3, applying sparsity-aware gating on top of ReLU experts further boosts interpretability.

## 6. Conclusion

This paper addresses the challenge of improving interpretability in LLMs by introducing MoE-X, a mixture-of-experts architecture designed for intrinsic transparency. Our key finding is that sparsity and width are essential for interpretability. By structuring MoE as wide and sparse MLP layers, we show that it naturally enhances interpretability. To further improve this, we use ReLU-based experts and sparsity-aware routing to reduce polysemanticity and create

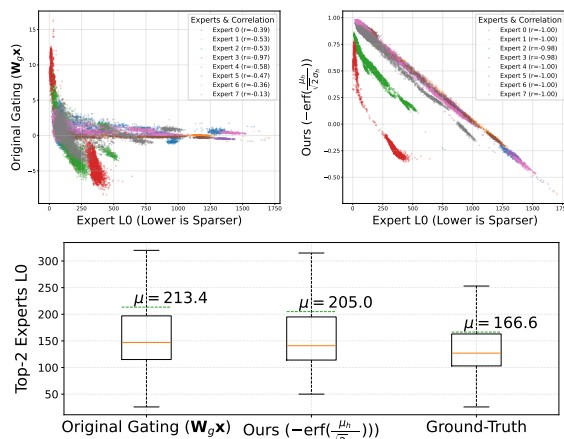

*Figure 9.* Comparison between TopK gating and our Sparsity routing. Our score identifies a more sparse set of experts.

sparser internal representations. Experiments on chess and language tasks demonstrate that MoE-X performs on par with dense Transformers while providing more interpretable outputs.

## Impact Statement

This paper introduces MoE-X, a scalable and interpretable language model designed to promote trust and reliability in AI systems. By improving transparency with sparse activations and efficient routing, MoE-X helps make AI decisions easier to understand. It can be especially useful in fields like healthcare and education, where trust is critical. While there is some risk of misuse or bias, these can be addressed through careful and ethical use. Overall, this work aims to advance AI by providing a more transparent and reliable approach to large-scale models.

## Acknowledgements

This work is supported by the UKRI grant: Turing AI Fellowship EP/W002981/1.

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

In the appendix, we provide additional details to complement our paper. Section A explores how interpretability scores evolve during training and across different model layers. Section B defines the metrics used to assess model interpretability. Section D describes the auto-interpretability experiment setup and presents newly identified interpretable features in the MoE-X small model. Section E provides a full derivation of how Sparse MoE can be formulated as an MLP layer and our sparse-aware gating. Finally, Section F details the model training configurations.

## A. Interpretability Dynamics

We evaluate two types of interpretability dynamics in a language model trained on chess. First, we study how interpretability evolves over the number of training steps. Second, we examine how interpretability varies across different layers of the language model. For our experiments, we use an 8-layer GPT-style model with configuration ($\alpha = 4, d = 512$).

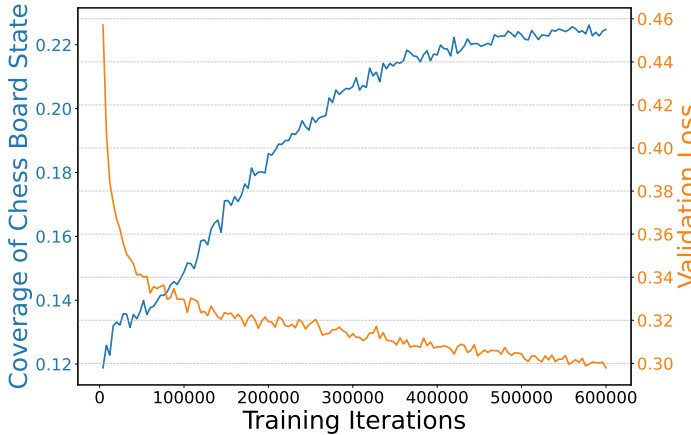

*Figure 10.* Dynamics of BSP Coverage Score and Validation Loss over Training Steps.

**Training Iterations & Interpretability.** As shown in Figure 10, the interpretability coverage score generally increases as training progresses. However, the trend does not exactly mirror the validation loss. Even when the validation loss plateaus, the coverage score continues to increase, indicating ongoing improvements in interpretability.

This observation motivates our decision to upcycle model weights from a dense model, following (Komatsuzaki et al., 2022). We find that longer training leads to higher interpretability scores. However, in MoE models, training is inherently less efficient per expert. Given a total of $T$ iterations, each expert in a MoE model is only activated and trained for $Tk/M$ iterations, where $k$ is the number of selected experts per step. As a result, each expert is effectively under-trained compared to a dense model, making direct interpretability comparisons unfair.

To validate this and assess the impact of weight upcycling, we conduct an experiment comparing different training strategies. Specifically, we train a MoE both from scratch and with upcycled dense weights while also continuing the training of a dense model for the same number of iterations.

The results, shown in Table 4, indicate that upcycling significantly improves interpretability. The MoE trained from scratch achieves a higher coverage score than the dense model, but its reconstruction performance lags behind. In contrast, the upcycled MoE not only outperforms the dense models in interpretability but also shows the best reconstruction score, demonstrating the benefits of leveraging pre-trained dense weights.

| Method | Coverage | Reconstruction |
|---|---|---|
| Dense | 0.356 | 0.608 |
| Dense (Continued Training) | 0.377 | 0.674 |
| MoE-X (Scratch) | 0.398 | 0.657 |
| MoE-X (Up-cycle) | **0.428** | **0.840** |

*Table 4.* Comparison of interpretability scores for different training methods.

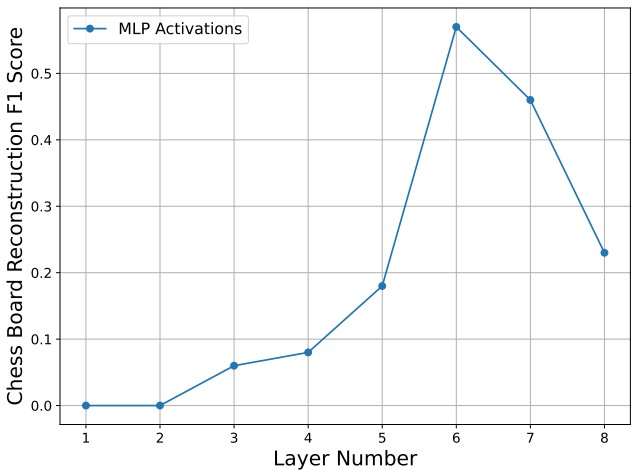

*Figure 11.* BSP Reconstruction Score at different language model layers.

**Layer number & Interpretability.** In Figure 11, we show the BSP Reconstruction Score across different layers of the language model. The interpretability score increases initially, peaks at layer 6, and then decreases. Based on this observation, we evaluate layer 6 in our 8-layer transformer. Similarly, we select moderate layers (e.g., layer 8 for a 12-layer model and layer 16 for a 24-layer model) for other model depths in natural language experiments.

## B. Metrics Definitions

### B.1. Board State Properties in Chess

We define a board state property (BSP) as a function $g : \{\text{game board}\} \rightarrow \{0, 1\}$, which evaluates specific characteristics of a board state. In this work, we focus on interpretable classes of BSPs that capture fundamental game properties.

One such class, $\mathcal{G}_{\text{board state}}$, includes BSPs that determine whether a specific piece is present at a given board square. Chess use an $8 \times 8$ board. In chess, we consider the full board for all twelve distinct piece types (e.g., white king, white queen, ..., black king), resulting in a total of $8 \times 8 \times 12$ BSPs.

### B.2. Coverage

The **coverage** metric evaluates how well the features learned by a Sparse Autoencoder (SAE) align with a given set of Board State Properties (BSPs). Let $G$ be a collection of BSPs, and let $\{f_i\}$ denote the set of features learned by the SAE. For each feature $f_i$, we define a binary classifier $\phi_{f_i, t}$ based on a threshold $t \in [0, 1]$:

$$\phi_{f_i, t}(x) = \mathbb{I}[f_i(x) > t \cdot f_i^{\max}],$$

where:

- $f_i(x)$ is the activation of feature $f_i$ for input $x$,
- $f_i^{\max} = \max_{x \sim D} f_i(x)$ is the maximum activation of $f_i$ over the dataset $D$,
- $\mathbb{I}[\cdot]$ is the indicator function, which outputs 1 if the condition is true and 0 otherwise.

For a given BSP $g \in G$, the F1-score of $\phi_{f_i, t}$ as a classifier for $g$ is denoted by $F_1(\phi_{f_i, t}; g)$. The **coverage** of the SAE with respect to $G$ is then defined as:

$$\text{Cov}(\{f_i\}, G) = \frac{1}{|G|} \sum_{g \in G} \max_t \max_{f_i} F_1(\phi_{f_i, t}; g).$$

In words, for each BSP $g$, we select the feature $f_i$ and threshold $t$ that maximize the F1-score for classifying $g$. The coverage score is the average of these maximal F1-scores across all BSPs in $G$. A coverage score of 1 indicates that the SAE has at least one feature that perfectly classifies every BSP in $G$.

### B.3. Board Reconstruction

The **board reconstruction** metric measures the ability of an SAE to recover the complete state of a chessboard from its feature activations in a human-interpretable way. Let $G$ be a set of BSPs, and let $\{f_i\}$ denote the set of SAE features. For each feature $f_i$, we identify the subset of BSPs $g \in G$ for which $\phi_{f_i,t}$ is a high-precision classifier (precision $\geq 0.95$) on a training dataset $D_{\text{train}}$.

For a given activation $x$, the predicted state of a BSP $g \in G$ is determined by the rule:

$$P_g(\{f_i(x)\}) = \begin{cases} 1, & \text{if } \phi_{f_i,t}(x) = 1 \text{ for any } f_i \text{ that is high-precision for } g \text{ on } D_{\text{train}}, \\ 0, & \text{otherwise.} \end{cases}$$

The full predicted board state is represented as $P(\{f_i(x)\}) = \{P_g(\{f_i(x)\})\}_{g \in G}$, which contains predictions for all 64 squares of the chessboard. The quality of the reconstruction is evaluated using the F1-score of the predicted board state $P(\{f_i(x)\})$ compared to the true board state $b$, denoted as $F_1(P(\{f_i(x)\}); b)$.

The **board reconstruction** score is then computed as the average F1-score over all board states in a test dataset $D_{\text{test}}$:

$$\text{Rec}(\{f_i\}, D_{\text{test}}) = \frac{1}{|D_{\text{test}}|} \sum_{x \in D_{\text{test}}} F_1(P(\{f_i(x)\}); b(x)),$$

where $b(x)$ is the true board state corresponding to activation $x$. This metric reflects how well the SAE's feature activations can be combined to reconstruct the full board state, emphasizing interpretability and precision.

## C. Routing Regularizes Sparsity

Our sparsity-aware gating significantly reduces expert sparsity. In a 2-out-of-8 MoE setup with top-k gating and ReLU experts, the $\ell_0$ norm of the experts is approximately 313. With our sparsity-aware gating, this value decreases to around 166. This demonstrates that sparsity-aware gating greatly enforces sparsity.

## D. Auto-Interpretability

We conduct an auto-interpretability experiment following the approach described in (Paulo et al., 2024).

For a MLP layer, We collected latent activations from the MLP over a 10M token sample of RedPajama-v2[4](Weber et al., 2024). The activations are gathered from batches of 256 tokens, each starting with a beginning-of-sentence (BOS) token.

To interpret these activations, we use Llama 3.1 70B Instruct as the explainer model. It is presented with 20 activating examples, each consisting of 32 tokens, where the activating tokens can appear at any position. These examples are randomly selected from a larger dataset to ensure diversity in activation patterns.

For evaluation, we use the detection score, where a scorer model identifies which sequences activate a given latent based on an interpretation. In this setup, the model is shown five examples at a time, each with an independent probability of activating the latent, regardless of the others. Each latent is evaluated using 100 activating and 100 non-activating examples. The activating examples are selected through stratified sampling, ensuring that 10 examples are drawn from each of the 10 deciles of the activation distribution.

**Example.** Besides the sample showed in the main paper, we show more results in the Table 5

---

[4]https://huggingface.co/datasets/togethercomputer/RedPajama-Data-V2

| Auto-Interp Meaning | Location | Example |
|---|---|---|
| Time of day in expressions | Expert 2, #457 | "We went for a walk in the **evening**." |
| | | "The meeting is scheduled for **afternoon**." |
| | | "She always exercises in the **morning**." |
| Abbreviations with dots | Expert 5, #89 | "She explained the concept using **e.g.** as an example." |
| | | "You must submit all forms by Friday, **i.e.**, tomorrow." |
| | | "Common abbreviations include **a.m.** and **p.m.** for time." |
| Capitals at the start of acronyms | Expert 6, #1601 | "The **NASA** mission was successful." |
| | | "The company developed cutting-edge **AI** systems." |
| | | "Students use **PDF** documents for submissions." |
| Ordinal numbers in sentences | Expert 3, #412 | "He finished in **1st** place." |
| Hyphenated compound words | Expert 2, #187 | "This is a **well-being** initiative." |
| Currency symbols preceding numbers | Expert 1, #273 | "The total cost was **$100**." |
| Parentheses around numbers or letters | Expert 6, #91 | "Refer to section **(a)** for details." |
| Ellipsis usage | Expert 0, #55 | "He paused and said, **...** I'll think about it." |
| Measurements followed by units | Expert 0, #384 | "The box weighs 5 **kg**." |
| Dates in numeric formats | Expert 7, #401 | "The deadline is **2025-01-29**." |
| Repeated punctuation marks | Expert 2, #1128 | "What is happening **???**" |
| Hashtags in text | Expert 4, #340 | "Follow the trend at **#trending**." |
| Uppercase words for emphasis | Expert 4, #278 | "The sign read, **STOP** immediately!" |
| Colon in timestamps | Expert 3, #521 | "The train arrives at 12**:**30." |
| Contractions with apostrophes | Expert 6, #189 | "I **can't** do this alone." |

*Table 5.* Sampled Activated Tokens and Contexts for Neurons in MoE-X Small. The meanings are identified by the Auto-interp process.

# E. Derivation

### E.1. MoE Layer as a Sparse MLP

In this section, we demonstrate how a Mixture-of-Experts (MoE) layer can be reformulated as a large and sparse Multi-Layer Perceptron (MLP).

The output of the MoE layer is expressed as a weighted sum of the expert outputs:

$$\hat{\mathbf{y}} = \sum_{j=1}^{M} \omega_j f_j(\mathbf{x}; \theta_j), \tag{7}$$

$$= \sum_{j=1}^{M} \omega_j \left( \mathbf{W}_{\text{dec}}^{(j)} \sigma(\mathbf{W}_{\text{enc}}^{(j)} \mathbf{x}) \right) \tag{8}$$

$$= \sum_{j=1}^{M} \mathbf{W}_{\text{dec}}^{(j)} \left( \omega_j \sigma(\mathbf{W}_{\text{enc}}^{(j)} \mathbf{x}) \right) \tag{9}$$

$$= \sum_{i=1}^{M} \mathbf{W}_{\text{dec}}^{(j)} \left( \omega_j \mathbf{z}^{(j)} \right) \tag{10}$$

where $\omega_j$ is the gating weight for the $j$-th expert, and $\mathbf{z}^{(j)} = \sigma(\mathbf{W}_{\text{enc}}^{(j)} \mathbf{x})$ is the hidden representation after the activation function $\sigma$ in the $j$-th expert. Since $\omega_j$ is a scalar, it can be factored out before multiplication with $\mathbf{W}_{\text{dec}}^{(j)}$.

To simplify this representation, we define a "mega-decoder" by concatenating all expert decoder matrices:

$$\mathbf{W}_{\text{dec}} = \text{concat}([\mathbf{W}_{\text{dec}}^{(1)}, \dots, \mathbf{W}_{\text{dec}}^{(M)}]) \in \mathbb{R}^{MD \times d}$$

Similarly, we concatenate the scaled hidden representations of all experts:

$$\mathbf{z} = \text{concat}([\omega_1 \mathbf{z}^{(1)}, \dots, \omega_M \mathbf{z}^{(M)}]) \in \mathbb{R}^{MD},$$

With these definitions, the MoE output can be reformulated as:

$$\hat{\mathbf{y}} = \mathbf{W}_{\text{dec}} \mathbf{z}.$$

This reformulation demonstrates that an MoE layer is equivalent to a wide and sparse MLP, where sparsity is induced by the selective activation of only a subset of experts for a given input.

Interestingly, a similar derivation is mentioned in (Liu et al., 2023c). However, their work focuses on building efficient and sparse neural networks, while ours emphasizes interpretability.

### E.2. Sparsity-Aware Gating

Suppose we have $E$ experts, each accosiated $\mathbf{W}_{\text{enc}} = \begin{bmatrix} w_{1,1} & \dots & w_{1,d} \\ \dots & \dots & \dots \\ w_{D,1} & \dots & w_{D,d} \end{bmatrix} \in \mathbb{R}^{D \times d}$ and input $\mathbf{x} = \begin{bmatrix} x_1 \\ \dots \\ x_d \end{bmatrix} \in \mathbb{R}^d$. The hidden activation is

$$\mathbf{z} = \mathbf{W}_{\text{enc}} \mathbf{x} = \begin{bmatrix} \sum_i w_{1,i} x_i \\ \dots \\ \sum_i w_{D,i} x_i \end{bmatrix} = \begin{bmatrix} z_1 \\ \dots \\ z_D \end{bmatrix} \tag{11}$$

If we assume that each rows of $\mathbf{W}_{\text{enc}}$ is i.i.d from a Gaussian distribution, $\{w_{j,i}\}_{j=1}^D \sim \mathcal{N}(\mu_i, \sigma_i^2)$. Then we can see each element of $\mathbf{z}$ from a mixture of gaussian distribution

$$z_j = \sum_i w_{j,i} x_i \sim \mathcal{N}(\mu_z, \sigma_z^2) = \mathcal{N}(\sum_i \mu_i x_i, \sum_i \sigma_i^2 x_i^2) \tag{12}$$

If we apply a `ReLU` function on top of this hidden, the probability that $z_j$ is positive is $P(z_j > 0)$. $P(z_j > 0)$ directly corresponds to the sparsity of `ReLU`$(z_j)$. Given the Gaussian assumption,

$$P(z_j > 0) = 1 - \Phi(-\frac{\mu_z}{\sigma_z}) = \Phi(\frac{\mu_z}{\sigma_z}) = \frac{1}{\sqrt{2\pi}} \int_{-\infty}^{\frac{\mu_z}{\sigma_z}} \exp\left\{ -\frac{u^2}{2} \right\} du \tag{13}$$

Where $\Phi(z) = P(Z \leq z)$ is the CDF of the normal distribution. In practice, a common closed-form approximation for the CDF $\Phi$ is

$$\Phi(z) \approx \frac{1}{2}(1 + \text{erf}(z/\sqrt{2})) \tag{14}$$

The larger the sparsity, the less non-zero values, and the $P(z_j > 0)$ gets smaller. Because we want to select the expert with the largest sparsity

Therefore, we select the experts with the smallest $P(z_j > 0)$

$$\arg\max_j \left[ \text{Sparsity}(\mathbf{W}_{\text{enc}}) \right] = \arg\min_j \left[ P(z_j > 0) \right] \tag{15}$$

In terms of the Gaussian CDF

$$\arg\min_j \Phi(\frac{\mu_z}{\sigma_z}) = \arg\max_j -\frac{1}{2}\left(1 + \text{erf}(\frac{\mu_z}{\sqrt{2}\sigma_z})\right) = \arg\max_j -\text{erf}(\frac{\mu_z}{\sqrt{2}\sigma_z}) \tag{16}$$

We use this as the gating function for mixture-of-expert

$$\mathbf{w} = g(\mathbf{x}; \phi) = \text{Softmax}(\text{TopK}(-\text{erf}(\frac{\mu_z}{\sqrt{2}\sigma_z}))) \tag{17}$$

## F. Training Details

The training configuration and hyperparameters are presented in Table 6 and Table 7.

*Table 6.* MoE & GPT-2 Training Configuration for Chess Dataset.

| Parameter | Value |
| --- | --- |
| Num layer | 8 |
| Num head | 8 |
| Num embd | 512 |
| dropout | 0.0 |
| Init learning rate | 3e-4 |
| Min lr | 3e-5 |
| Lr warmup iters | 2000 |
| Max iters | 600000 |
| optimizer | AdamW |
| batch size | 100 |
| context len | 1023 |
| Num experts | 8 |
| Num experts per Token | 2 |
| grad_clip | 1.0 |

*Table 7.* MoE & GPT-2 Small Training Configuration for FineWeb Language Tasks.

| Names | Small | Medium |
| --- | --- | --- |
| Num layer | 12 | 24 |
| Num head | 12 | 16 |
| Num embd | 768 | 1024 |
| dropout | 0.0 | 0.0 |
| Init learning rate | 3e-4 | 3e-4 |
| Min lr | 3e-5 | 3e-5 |
| Lr warmup iters | 5000 | 5000 |
| Max iters | 100000 | 100000 |
| optimizer | AdamW | AdamW |
| batch size | 320 | 320 |
| context len | 1024 | 1024 |
| Num experts | 8 | 8 |
| Num experts per Token | 2 | 2 |
| grad_clip | 1.0 | 1.0 |

**Load Balance Loss.** To ensure a balanced distribution of tokens across experts, we use an auxiliary loss borrow from (Fedus et al., 2022). This auxiliary loss is added to the total model loss during training.

Given $N$ experts indexed by $i = 1$ to $N$ and a batch $B$ containing $T$ tokens, the auxiliary loss is defined as the scaled dot product between the token distribution vector $\mathbf{f}$ and the router probability vector $\mathbf{P}$:

$$\mathcal{L}_{\text{balance}} = \alpha \cdot N \sum_{i=1}^{N} f_i \cdot P_i \tag{18}$$

where $f_i$ represents the fraction of tokens assigned to expert $i$:

$$f_i = \frac{1}{T} \sum_{x \in B} \mathbb{I}\{\arg\max p(x) = i\} \tag{19}$$

and $P_i$ denotes the fraction of the router probability allocated to expert $i$:

$$P_i = \frac{1}{T} \sum_{x \in B} p_i(x). \tag{20}$$

Since we aim for uniform token routing across all $N$ experts, both $\mathbf{f}$ and $\mathbf{P}$ should ideally have values close to $1/N$. For all MoE model training in this paper, we set the load balancing weight to $\lambda = 0.001$.

