# OpenReview forum: "Mixture of Experts Made Intrinsically Interpretable"
_ICML.cc/2025/Conference — ICML 2025 poster_

### Official Review · Reviewer_nXvb · 2025-03-11

**Overall Recommendation:** 3

**Summary:**

The paper proposes a novel method for intrinsically interpretable LLMs, called MoE-X. It's goal is to achieve better interpretability than sparse autoencoder by leveraging mixture of experts and providing sparse explanation without polysemanticity of activations. To do so there is a proposal to use wide and sparse experts and a routing to properly handle the computations. The experiments are performed on Chess benchmark, and OpenWebText, WikiText103 and WikiText2. Perplexicity is used as a metric.

**Claims And Evidence:**

claims are presented clearly

**Essential References Not Discussed:**

all essentials works are cited. However, there is no discussion with alternative to MoE solutions based on prototypical parts. and comparison, e.g. with work of:

Xie, Sean, Soroush Vosoughi, and Saeed Hassanpour. "Proto-lm: A prototypical network-based framework for built-in interpretability in large language models." EMNLP (2023).

**Ethical Review Concerns:**

It is an XAI method for LLMs, and the usage of LLMs should be monitored.

**Ethical Review Flag:**

Flag this paper for an ethics review.

**Ethics Expertise Needed:**

["Legal Compliance (e.g., GDPR, copyright, terms of use)"]

**Experimental Designs Or Analyses:**

The metrics for interpretability is perplexity only, this is strongly limiting the work. Additionally, there is no user study showcasing that those explanations are better comprehensible by the users.

To fill the gap of metrix, authors should also consider reporting faithfulness score, and Simulatability. As well as conduct a user study showcasing their superiority.

**Methods And Evaluation Criteria:**

methods and evaluation criteria are presented clearly

**Other Comments Or Suggestions:**

I do not see a time to provide additional results in a author response time.

**Other Strengths And Weaknesses:**

well written

**Questions For Authors:**

Could you provide comparison of your method to Proto-lm?

Also can you provide evaluation results of other metrics widely used in XAI such as faithfullness?

I have read the rebuttal and increased my score accordingly. Good job for the authors to provide comparison! It was a challenging task given the time.

**Relation To Broader Scientific Literature:**

They are related works and broader aspects of the field correctly.

**Theoretical Claims:**

NA

---

> ### Author Rebuttal · Authors · 2025-03-30
>
> We sincerely thank R-nXvb for their valuable questions!
>
> `>>> Q1`**Evaluation Metrics**
>
> `>>> A1`The reviewer noted that `metrics for interpretability is perplexity only`. We appreciate the feedback, but would like to clarify two critical points
>
> 1. **Misconception About Perplexity**:  Perplexity is **not an interpretability metric**; it measures language modeling accuracy.
> 2. **Interpretability Metrics**:  Our work evaluates interpretability using **three distinct metrics**
>    - *Coverage/Reconstruction Score* (Table 1): Measures the alignment between the hidden neurons and human-defined chess states.
>    - *Auto-Interp Score* (Figure 8): Assesses the accuracy of predicting activations for unseen samples using extracted explanations.
>    - *Case Studies* (Figure 7): Provide qualitative validation of interpretability.
>   These metrics are commonly used in previous research.
>
> Additionally, both `R-K7C2` and `R-tfxC` acknowledged that our evaluation is reasonable. Based on this, we believe our evaluation is sufficient.
>
> `>>> Q2`**Comparing with Proto-lm [A]**
>
> `>>> A2`Really thank the reviewer for bring this great work! We will cite Proto-lm in our revision. While both methods share similar motivation, key differences make direct comparison challenging:
> 1. **Base Models**: Proto-lm focuses on *encoder-only models* (e.g., BERT), while MoE-X is designed for *decoder-only causal LLMs* like GPT.
> 2. **Tasks**: Proto-lm targets text classification (e.g., SST2), whereas MoE-X is for general language generation. So metrics like *Simulatability* cannot be directly applied to our model.
> 3. **Layers**. Proto-lm explains only the *final layer's embeddings*, while MoE-X provides interpretability across all layers.
> 4. **Interpretability Categorization**. Proto-lm uses **prototype-based explanations** for local, per-sample explaination. On the other hand, MoE-X focuses on **mechanical interpretability**, offering a global explanation of how the network functions by identifying the meaning of individual neuron.
>
> `>>> Q3`**User Study**
>
> `>>> A3`Thanks for the suggestion! As requested, we conduct a blinded human evaluation to assess neuron monosemanticity in MoE-X compared to GPT-2 and GPT-2 + SAE.
>
> Specifically, 5 raters evaluate 20 random features from each model. Based on the top 5 activating samples, raters classified each feature as: `Yes` (clearly monosemantic), `Maybe` (partially interpretable), `No` (not monosemantic).
>
> Results showed MoE-X’s features were most interpretable, with 70% (_14/20_) labeled as `Yes`.
>
> |Model|Yes|Maybe|No|
> |-|-|-|-|
> |GPT-2|8|4|8|
> |GPT-2 + SAE|12|3|5|
> |MoE-X|14|3|3
>
> `>>> Q4`**Faithfulness and Simulatability Score**
>
> `>>> A4`As noted in `Q2`, directly comparing *Faithfulness* and *Simulatability* is not feasible due to differences in model, task, and setup. However, as suggested, we retrained the model and defined new evaluation metrics to make this comparison possible.
>
> Specifically, we fine-tune MoE-X-Medium on SST2 using the last token for classification. We report the interpretability on the final layer. For Proto-lm, scores are **extracted from the raw figure in its paper**, which may have minor discrepancies. This provides *reasonable, though not exact*, comparability for interpretability analysis.
>
> **Faithfulness**
>
> To evaluate faithfulness, we define two metrics inspired by [A].
> -   **Comprehensiveness (Comp)**: Measures the decrease in model confidence when top $k$% activated neurons are removed.
> -   **Sufficiency (Suff)**: Measures the change in confidence when only top $k$% activated neurons are preserved.
>
> We do evaluations on the SST2, with with $k\in$ {1,5,10, 20, 50}.
>
> **Results**:  The table shows *Comp* and *Suff* scores. Notably, for $k\geq 5$%, MoE-X **achieves perfect scores** (Comp=100%, Suff=0%) due to its extreme sparsity: MoE-X activates <100 neurons out of 8,192 in a layer. This ensures faithful explanations, outperforming Proto-lm.
>
> |k%|Comp&#8593;||Suff&#8595;||
> |-|-|-|-|-|
> ||Proto-lm|MoE-X  |Proto-lm|MoE-X |
> |1|69|**87**|36|**9**|
> |5|71|**100**|31|**0**|
> |10|81|**100**|27|**0**|
> |20|85|**100**|35|**0**|
> |50|88|**100**|50|**0**|
>
> **Simulatability**
>
> After training MoE-X on SST-2, we identify each neuron's concept and select the top 5 activated concepts for each sample. Following the evaluation setup in [A], 3 evaluators categorize 50 SST-2 questions based on these concepts.
>
> **Results**: MoE-X performs slightly worse than Proto-LM but outperforms other methods in [A]. Note that differences in question selection and human judgment make results not directly comparable to [A].
>
> |Method|SST-2|
> |-|-|
> |Random|42.3%|
> |LIME |87.3%|
> |Integrated Gradient|84.7%|
> |Proto-lm|**90.0%**|
> |**MoE-X**|88.7%|
>
> > Special Note: The detection score in paper is a Simulatability score. It predicts activation patterns (not classes) using an LLM, not humans.
>
> [A] "Proto-lm: A prototypical network-based framework for built-in interpretability in large language models." EMNLP (2023).

---

### Official Review · Reviewer_tfxC · 2025-03-12

**Overall Recommendation:** 4

**Summary:**

The paper presents MoE-X, a novel Mixture-of-Experts (MoE) architecture designed to enhance the interpretability of large language models (LLMs) while maintaining competitive performance. The authors explore the challenge of polysemanticity in neurons and its relationship to the model's architecture. They address this through architectural modifications that promote sparsity and width in the network. Key contributions include redesigning the MoE layer as a wide, sparse MLP with ReLU experts and sparsity-aware routing. The paper demonstrates through experiments on chess and natural language tasks that MoE-X achieves performance comparable to dense models while significantly improving interpretability metrics.

**Claims And Evidence:**

The claims about improved interpretability and maintained performance are supported by experimental results on chess and language tasks. The authors provide evidence through quantitative metrics (perplexity, BSP coverage score) and qualitative analysis (t-SNE visualizations, auto-interpretability examples). The discussion of architectural factors influencing interpretability is well-supported by preliminary studies and ablation analyses. However, the claim that MoE-X completely eliminates polysemanticity might be overstated, as some level of polysemanticity is inherent in neural networks, especially in language models where words have multiple meanings in different contexts

**Essential References Not Discussed:**

No.

**Experimental Designs Or Analyses:**

The experimental designs appear sound. The authors compare MoE-X against several baselines (dense models, other MoE variants) and use appropriate metrics for both performance and interpretability. The ablation studies help isolate the contributions of different components.

**Methods And Evaluation Criteria:**

The proposed methods (ReLU experts, sparsity-aware routing) and evaluation criteria (BSP coverage score, reconstruction score, detection accuracy) make sense for the problem of improving interpretability in language models. The chess dataset provides a clear ground truth for evaluating interpretability, and the natural language experiments use standard benchmarks.

**Other Comments Or Suggestions:**

No.

**Other Strengths And Weaknesses:**

It will make this paper more convincing if the feature streering experiments are involved.

**Questions For Authors:**

I would expect some degree of conflict between interpretability and performance. What specific strategies did you employ to balance interpretability gains against potential loss in performance?

**Relation To Broader Scientific Literature:**

The paper builds on prior work in mechanistic interpretability, sparse autoencoders, and MoE architectures.

The paper could benefit from citing recent work on differentiable MoE routing mechanisms, such as ReMoE[1], which also uses ReLU-based routing but focuses on differentiability and load balancing.

[1] Wang, Ziteng, Jianfei Chen, and Jun Zhu. "ReMoE: Fully Differentiable Mixture-of-Experts with ReLU Routing." arXiv preprint arXiv:2412.14711 (2024).

**Theoretical Claims:**

N/A

---

> ### Author Rebuttal · Authors · 2025-03-30
>
> We sincerely appreciate Reviewer tfxC's suggestions. We have carefully incorporated them into the revised manuscript.
>
> `>>> Q1`**Related Work on MoE**
>
> `>>> A1`We truly appreciate R-tfxC for bringing the ReMoE [1] paper on sparse MoE to our attention. We will definitely include a citation in the revised version.
>
> While both work utilize the properties of ReLU in MoE, there are key differences:
> 1. **Purpose of study**: ReMoE aims to make the routing function fully differentiable, while our focus is on enhancing the interpretability of the MoE.
> 2. **Position where ReLU used**: In our work, ReLU is applied within each expert, whereas ReMoE uses it in the gating function.
>
> `>>> Q2`**Feature Streering Experiments**
>
> `>>> A2`We truly appreciate the suggestion.
> While we cannot perform a full qualitative comparison within the rebuttal period, we provide an initial analysis by applying feature steering to MoE-X and evaluating the results.
>
> **Love − Hate Steering**
>
> Building on the **Activation Addition** method [A], we demonstrate how feature steering can influence model completions. Using MoE-X-S, we first extract feature vectors for the contrasting concepts `"Love"` and `"Hate"`.
>
> These vectors are then used to compute a steering adjustment
>  `v_Steered = v_Unsteered + coefficient * (v_"Love" − v_"Hate")`. We apply the steering at **MoE hidden feature at layer 6** with a coefficient of **5**.
>
> The table below compares unmodified model completions (left) with steered ones (right). The bolded text represents the prompt.
>
> | **Unsteered Completions** | **Steered Completions** |
> | :-- | :-- |
> | **I hate you because** you're spineless. You know what I hate? People who believe the only way to accomplish something is to go through the motions without real effort. Those who hesitate to take meaningful action are... | **I hate you because** you're an incredible person. The reason I'm here is that I truly care about you. I don't want to be distant anymore. I want to embrace the present and move forward with hope... |
>
> This demonstrates how targeted feature modifications can also steer model behavior in our MoE-X.
>
> > [A] Turner, Alexander Matt, et al. "Activation addition: Steering language models without optimization." _arXiv e-prints_ (2023): arXiv-2308.
>
> `>>> Q3`**Interpretability and Performance**
>
> `>>> A3`The review is absolutely right -- there is indeed a trade-off between interpretability and performance. If we enforce extreme sparsity (e.g., pushing the  $l_0$​  norm to zero), the model would have no activations and thus fail to learn.
>
> To address this, we use  **ReLU-based sparsity**  rather than hard constraints like $l_1$​ regularization (as in SAEs). This allows the model to  *adaptively learn sparse patterns*  while still maintaining strong performance. As a result, our approach preserves interpretability without sacrificing the model's ability to fit the data effectively.

---

### Official Review · Reviewer_K7C2 · 2025-03-13

**Overall Recommendation:** 4

**Summary:**

The paper introduces MoE-X, a Mixture-of-Experts (MoE) language model designed to be intrinsically interpretable. This is different from the recent trend of using Sparse Autoencoders to interpret the model representations at post-doc. The proposed method addresses the challenge of polysemanticity in large language models (LLMs) representations, where individual neurons encode multiple unrelated concepts, making post-hoc interpretability difficult. The authors propose a novel architecture by leveraging MoE’s sparsity and width to encourage disentangled representations. Extensive experiments on both chess and language tasks prove the effectiveness of the proposed method.

**Claims And Evidence:**

The major claims made by the paper are:
1. MLP Hidden Size: Larger hidden states result in better interpretability.
2. Sparsity of Hidden Activations: Lower numbers of nonzero neurons lead to more interpretable representations.

Overall, the claims are well-supported by experiments.

**Essential References Not Discussed:**

The paper covers the most relevant references.

**Experimental Designs Or Analyses:**

As in *Methods and Evaluation Criteria,* the experimental setup is reasonable, and the results support the claims.

**Methods And Evaluation Criteria:**

The proposed evaluation methods are reasonable.
1. Chess dataset: Used as a structured benchmark with board state properties as a ground truth, which is appropriate.
2. Automated interpretability pipeline on natural language tasks: This is a standard method used in other related works.

**Other Comments Or Suggestions:**

Is the formatting a big issue? Since this paper does not use a template that has line numbers and indicating under review in ICML.

**Other Strengths And Weaknesses:**

Strengths:
1. The paper is well-written and easy to follow.
2. The proposed method is novel and technically sound.

Weaknesses:
1. The paper would benefit from additional application scenarios, such as medical NLP benchmarks. And also human evaluations.
2. It would be more rigorous to validate the Gaussian assumption in sparsity-aware routing via experiments.
3. It would be more interesting to see if we can leverage the proposed method to steer the model behaviors or correct wrong/biased predictions.

**Questions For Authors:**

Minor: Will increasing the number of experts significantly impact interpretability?

**Relation To Broader Scientific Literature:**

Different from the recent trend of using Sparse Autoencoders to interpret the LLM representations at post-doc, the authors borrow ideas from MoE to achieve intrinsically interpretable LLM, which is novel.

**Theoretical Claims:**

The theoretical claims focus on how MoE can be reformulated as a sparse MLP and how routing can be modified to enforce sparsity. The derivations appear correct.

---

> ### Author Rebuttal · Authors · 2025-03-30
>
> We truly thank the R-K7C2 for the nice comments.
>
> `>>> Q1`**Additional Application Scenarios**
>
> `>>> A1`We sincerely appreciate the suggestion! Expanding to the medical domain is valuable, but a key challenge is finding a benchmark that **evaluates both interpretability and performance**. While many medical datasets focus on performance, few assess interpretability, highlighting the need for a dedicated benchmark.
>
> As a first step, we tested the small MoE-X model (8×124M) on MMLU’s medical subset using 5-shot prompting, comparing it to a similarly sized dense model (GPT-Neo 125M).
>
> | Metric                 | GPT-Neo 125M |  MoE-X-S (8×124M) |
> |------------------------|--------------|------------------|
> | MMLU (5-shot)         | 26.0        | **29.3**            |
>
> While MoE-X-S performs good, we’d need more research to evaluate how interpretable it is in specialized domains.
>
> `>>> Q2`**Human Evaluation**
>
> `>>> A2`Thanks for the suggestion! We conducted a blinded human evaluation to assess neuron monosemanticity in MoE-X compared to baseline models (GPT-2 and GPT-2 with SAE).
>
> Following GatedSAE’s approach, 5 raters evaluated 20 random features from each model, presented in random order. Each rater evaluated 12 features (20 × 3 / 5). For each feature, raters were shown the top 5 activating samples. Raters classified each feature as: `Yes` (clearly monosemantic), `Maybe` (partially interpretable), `No` (not monosemantic).
>
> | Model | Yes | Maybe | No|
> |--|--|--|--|
> |GPT-2|8|4|8|
> |GPT-2 + SAE| 12 |3|5|
> |MoE-X | 14|3|3
>
> MoE-X’s features were rated as ​most interpretable, with 70% (_14/20_) labeled `Yes`. While this is a small-scale study, it provides a promising preliminary insight.
>
> `>>> Q3`**Validate the Gaussian assumption**
>
> `>>> A3`Very luckily, a recent arXiv paper [A] has conducted a formal study on investigates the distribution of weights in LLMs. The authors find that the **weights of all major open-source LLMs closely follow a Gaussian distribution**, including LLaMA, Qwen, and the Vicuna family. They provide both statistical evidence and theoretical explanations for this phenomenon. We will cite this work in the revised version!
>
> [A] Unveiling the Mystery of Weight in Large Foundation Models: Gaussian Distribution Never Fades https://arxiv.org/abs/2501.10661
>
>
> `>>> Q4`**MoE Feature Steering**
>
> `>>> A4`We truly appreciate the suggestion.
> While we cannot perform a full qualitative comparison within the rebuttal period, we provide an initial analysis by applying feature steering to MoE-X and include the results below.
>
> **Love − Hate Steering**
>
> Building on the **Activation Addition** method [B], we demonstrate how feature steering can influence model completions. Using MoE-X-S, we first extract feature vectors for the contrasting concepts `"Love"` and `"Hate"`.
>
> These vectors are then used to compute a steering adjustment
>  `v_Steered = v_Unsteered + coefficient * (v_"Love" − v_"Hate")`. We apply the steering at **MoE hidden feature at layer 6** with a coefficient of **5**.
>
> The table below compares unmodified model completions (left) with steered ones (right). The bolded text represents the prompt.
>
> | **Unsteered Completions** | **Steered Completions** |
> | :-- | :-- |
> | **I hate you because** you're spineless. know what I hate? People who believe the only way to accomplish something is to go through the motions without real effort. Those who hesitate to take meaningful action are... | **I hate you because** you're an incredible person. The reason I'm here is that I truly care about you. I don't want to be distant anymore. I want to embrace the present and move forward with hope... |
>
> This demonstrates how feature addition can also steer model behavior in our MoE-X.
>
> [B] Turner, Alexander Matt, et al. "Activation addition: Steering language models without optimization." _arXiv e-prints_ (2023): arXiv-2308.
>
> `>>> Q5`**Formatting issue**
>
> `>>> A5`We sincerely apologize for the formatting and line number issues! We will make sure to correct them in the revision.
>
> `>>> Q6`**Impact of number of experts**
>
> `>>> A6`Thanks for the insightful question! As suggested, we explore two ways to increase the number of experts, and both show promising improvements:
>
> 1.  **Activating More Experts (with a fixed total number)**: In fact, this has been shown in `Fig 5` in the paper, which we varying the number of activated experts
> $k \in \{1,2,4\}$. It shows that increasing the number of activated experts leads to better interpretability.
>
> 2) **Expanding Total Experts (keeping the number of activated experts constant)**: We increased the total experts from 8 to 16. This improved interpretability. However, the improvements were modest compared to increasing activated experts, likely because the per-inference cost remained the same.
>
> |Model |Val Loss&#8595;|Coverage&#8593;|Reconstruction&#8593;|
> |--|--|--|--|
> MoE-X (2 from 8, in paper) | 0.211 | 0.428| 0.840|
> MoE-X (2 from 16) | **0.207** | **0.442** | **0.853**|

---

> > ### Comment · Reviewer_K7C2 · 2025-04-02
> >
> > Thank you to the authors for the detailed responses to my questions and the interesting additional results. The arXiv paper for Gaussian Distribution is also insightful. I am happy to maintain my score as Accept.

---

### Official Review · Reviewer_a7ZR · 2025-03-14

**Overall Recommendation:** 3

**Summary:**

This paper proposes a variant of an MoE architecture called MoE-X that makes design decisions that boosts the mechanistic interpretability of the model, while largely preserving quality. The authors authors motivate this with a preliminary study on the importance of MLP hidden size and sparsity of hidden activations for interpretability, using Chess board state prediction as an example. They then draw connections to why MoEs are essentially a special case of a large, sparse MLPs. They then design MoE-X which extends the MoE with various design decisions to boost interpretability: notably, the use of ReLU activation, and a novel sparsity-aware routing scheme. They show that on chess, MoE-X is more interpretable, than dense, MoE, and post-hoc interpretability baselines (SAE). The same holds true when pretraining the model and baselines over natural language (FineWeb): MoE-X is more interpretable than GPT-2 quantitatively, while achieving similar perplexity evals to normal MoE models. Qualitatively MoE-X activations mirror that typically obtained via more involved automated interpretability detection methods.

**Claims And Evidence:**

The claims in this work are sound and are supported by clear and convincing evidence. The experimental settings certainly have limitations, but the authors present the results clearly and straightforwardly, and do not overclaim. For the most part, both the qualitative examples, and quantitative results provide convincing evidence that MoE-X exhibits improved interpretability over baselines while maintaining quality.

**Essential References Not Discussed:**

N/A

**Experimental Designs Or Analyses:**

Apart from the limited evaluations noted above, my primary concern with this paper lies in the comparison between GPT-2 and MoE-X in this paper. While MoE-X and Switch are parameterized identically, and thus are comparable, the comparison against GPT-2 is a bit unfair as the small and medium parameterizations for MoE is slightly larger in terms of activated params / FLOPs than the dense baseline. Although in absolute terms the difference is not that large, a small scales like those studied in this paper 124M vs 180M or 354M vs 555M is a meaningful relative difference of ~50%. This could contribute to both the quality gains and the increased interpretability (extra MLP params) when compared against the Dense baseline.

It seems then the true baseline for MoE-X is more similar to Switch Transformer. On chess this comparison is shown well, but Figure 8 does not show this comparison w.r.t. interpretability for Switch Transformer. This somewhat calls into question the validity of some of the comparisons made in the work. In general, the comparison to SAE could be applied more rigorously -- how does this method compare to SAE as applied to MoEs?

**Methods And Evaluation Criteria:**

Evaluation criteria and methodology in this work is on the smaller scale end, focusing on a toy example (chess) and small scale language modeling, with evaluation only on perplexity.

If the intention of this work is to spark interest in the field of interpretable model design, this may be sufficient. However, most architecture proposals typically strive to validate that the quality of the model is either neutral or better across a wide array of settings, in order to prove to the reader that the architecture is worth adopting. This paper would be significantly stronger if it showed that on common pretraining evaluations, such as few-shot learning, the proposed architecture did not regress over an identical MoE parameterization without these modifications. It is difficult to judge how realistically this model could be used via Table 1 and 2 in this work, which are quite limited evaluation of LM performance.

**Other Comments Or Suggestions:**

Paper contains multiple typos, such as in the intro "Waswani et al." instead of "Vaswani", and in the section "Study II: Activation Sparsity". Stylization of certain methods could be improved, such as "AdamW" instead of "Adamw".

**Other Strengths And Weaknesses:**

Strengths:
- The design of sparsity-aware routing for interpretability is novel, interesting, and appears effective. It honestly could be worth a rigorous study / paper on its own.
- It's positioning / contribution to the field is meaningful and valuable as noted above.

Weaknesses:
- Presentation generally needs to be polished.

**Questions For Authors:**

Do you have a Sparse Transformer baseline for Figure 8? How does it compare against MoE-X?

**Relation To Broader Scientific Literature:**

This paper makes an interesting contribution to the field as it explores how to make an existing state-of-the-art architecture (token-routed MoEs) more interpretable while maintaining its quality. This is in opposition to post-hoc interpretability techniques over such models, which might be suboptimal. Figure 5 provides compelling evidence this is the case. This is a compelling direction as if a modeling proposal is neutral but improves interpretability, then it could be adopted almost for free. This work's novel sparsity-aware routing seems to be a meaningful step towards this goal.

**Theoretical Claims:**

The paper doesn't make any proofs or theoretical claims. I am familiar with the theoretical connection between wide MLPs and MoEs presented in the motivation of the work, and their description is sound.

---

> ### Author Rebuttal · Authors · 2025-03-30
>
> We sincerely appreciate R-a7ZR's thoughtful comments and suggestions.
>
> `>>> Q1`**Evaluation Setup and Additional Experiments**
>
> `>>> A1`We truly appreciate the suggestion! As R-a7ZR mentioned, our primary focus is on **interpretable model design**. In line with this goal, we use small-scale datasets, guided by 3 key factors:
> 1. **Comparison with Prior Work**: We follow the same evaluation protocol used in prior studies (SAE on GPT-2 Small) for fair comparison.
> 2.  **Interpretability Evaluation**: We need a benchmark that evaluates both interpretability and performance. Small, controlled dataset like chess is good for this.
> 3. **Computational Constraints**: Our resources are limited to an 8×48G GPU server, which supports prototyping but not large-scale experiments.
>
> **Large-scale Evaluations**
>
> While large-scale training isn't feasible within the rebuttal period, we follow the reviewer’s suggestion and evaluate our trained checkpoints on additional standard few-shot benchmarks. The results are shown in the table below.
>
> Though preliminary (due to the model’s small size), our MoE-X model performs comparably to the Switch Transformer. We plan to scale up the model in future work.
>
> |Metric|GPT-Neo 125M|Switch-S (8×124M)|MoE-X-S (8×124M)|
> |-|-|-|-|
> |Avg.|25.8|**29.3**|28.6|
> |ARC (25-shot)|22.9|27.3|**28.3**|
> |HellaSwag (10-shot)|30.3|**34.1**|33.2|
> |MMLU (5-shot)|26.0|**30.4**|29.3|
> |TruthfulQA (0-shot)|45.6|**52.3**|49.5|
> |Winogrande (5-shot)|51.8|**54.6**|54.1|
> |GSM8K (5-shot)|0.3|**1.3**|1.2|
> |DROP (3-shot)|3.7|5.1|**6.3**|
>
> `>>> Q2`**Fair Comparison between GPT-2 and MoE-X**
>
> `>>> A2`Thank R-a7ZR for this thoughtful suggestion! We completely agree that a parameter-matched comparison is critical for fairness.
>
> As suggested, we re-train MoE-X Small with 1 activated expert (124M active) to match GPT-2 Small’s size. To speed up training, we fine-tune the old top-2 MoE-X, but activating only 1 expert per token. The training took ~8 hours. This top-1 MoE-X (124M) has the same active parameters as GPT-2 (124M).
>
> **Apples-to-Apples Comparison**:
> We report performance using perplexity. As shown below, MoE-X-S (1 expert active) outperforms GPT-2 Small across all datasets while using ​**identical active parameters**. But as expected not as good as the MoE-X-S (2 from 8×124M) with around 50% more parameters.
> |Model|OpenWeb (PPL)↓|LAMBADA (PPL)↓|WikiText103 (PPL)↓|WikiText2 (PPL)↓|
> |-|-|-|-|-|
> |GPT-2 Small|22.83|32.71|49.89|44.36|
> |MoE-X-S (1 activated)|21.36|30.92|47.17|44.07|
> |MoE-X-S (2 activated)|**19.42**|**28.11**|**43.80**|**42.58**|
>
> ​**Interpretability Gains Persist**:
> We evaluate interpretability using the *Detection Score* defined in the paper. Still, with the same activated parameters, MoE-X-S (1 from 8×124M) demonstrates a clear improvement in identifying more accurate concepts compared to GPT2-Small.
> |Quantiles/Accuracy|GPT2-Small|MoE-X-S (1 activated)|
> |-|-|-|
> |Not|**0.88**|0.84|
> |Q1|0.223|**0.265**|
> |Q2|0.233|**0.281**|
> |Q3|0.249|**0.301**|
> |Q4|0.230|**0.302**|
> |Q5|0.265|**0.342**|
> |Q6|0.311|**0.357**|
> |Q7|0.302|**0.369**|
> |Q8|0.315|**0.379**|
> |Q9|0.321|**0.388**|
> |Q10|0.354|**0.403**|
>
> These results highlight that MoE-X’s advantages stem not just from scale but from its *structured sparse design*. We deeply appreciate the suggestion and will include in the revised paper!
>
> `>>> Q3`**Switch Transformer in Figure 8**
>
> `>>> A3`Great suggestion! As suggested, we incorperate Switch Transformer into Figure 8. We re-run the auto-interp experiment for Switch-S and compare with MoE-X-S using the average *Detection Score*.
>
> While both models perform well, MoE-X-S consistently outperforms the standard MoE in interpretability across all quantiles.
>
> | Quantiles | Switch-S | MoE-X-S |
> |-|-|-|
> |Not|0.84|0.82|
> |Q1|0.305|**0.320**|
> |Q2|0.321|**0.331**|
> |Q3|0.330|**0.339**|
> |Q4|0.318|**0.330**|
> |Q5|0.356|**0.379**|
> |Q6|0.378|**0.396**|
> |Q7|0.436|**0.448**|
> |Q8|0.448|**0.461**|
> |Q9|0.515|**0.531**|
> |Q10|0.685|**0.699**|
>
>
>
> `>>> Q4`**SAE applied to MoEs**
>
> `>>> A4`That’s really an interesting direction! SAE can indeed be applied to MoEs, including MoE-X.
>
> Due to time constraints, we conduct a quick experiment by training SAE on MoE-X and Switch Transformer using the Chess dataset for post-training interpretability. The SAE is placed at `post-res` layer 6 with a hidden size of 4096.
>
> |Model|Val Loss|Coverage|Reconstruction|
> |-|-|-|-|
> Switch |0.212| 0.424| 0.734|
> Switch + SAE|0.222|0.430|0.824
> MoE-X|**0.211**|0.428|0.840|
> MoE-X + SAE|0.220| **0.433**|**0.857**|
>
> Three key observations:
> 1. SAE improves interpretability for MoE but increases validation loss.
> 2. Switch + SAE slightly outperforms MoE-X alone. But MoE-X + SAE fights back.
> 3. Improvements are modest due to SAE's hidden size matching MoE's channel width.
>
> `>>> Q5`**Typos and formating**
>
> `>>> A5`We sincerely thank the reviewer for the careful proofreading! We will correct the typos in the references and ensure proper formatting for terminology in the revision.

---

> > ### Comment · Reviewer_a7ZR · 2025-04-04
> >
> > Thank you authors for providing the additional results. It is inline with my expectations, but gives me additional confidence in the claims presented by the work. I would like to maintain my rating for now.

---

### Decision · Program_Chairs · 2025-05-01

**Decision:**

Accept (poster)

**Comment:**

In the paper, the authors propose MoE-X, an interpretable mixture of experts large language model, while maintaining competitive performance. All the reviewers are positive about the contributions of the papers, including: (1) the novelty of the proposed MoE-X; (2) the experiments are sufficient to support the claims about the interpretability of the MoE-X; (3) the paper is written and presented clearly. After the rebuttal, most of the concerns of the reviewers were addressed, and all the reviewers are happy with the current stage of the paper.

In my opinion, the contributions and originality of the proposed MoE-X are sufficient for acceptance at ICML. Therefore, I recommend accepting it in its current form. However, I encourage the authors to address the reviewers’ suggestions and integrate their feedback into the camera-ready version of their paper.